# Cytotoxicity Profiles and Neuroprotective Properties of the Novel Ifenprodil Analogues as Sigma Ligands

**DOI:** 10.3390/molecules28083431

**Published:** 2023-04-13

**Authors:** Daniele Zampieri, Antonella Calabretti, Maurizio Romano, Sara Fortuna, Simona Collina, Emanuele Amata, Maria Dichiara, Agostino Marrazzo, Maria Grazia Mamolo

**Affiliations:** 1Department of Chemical and Pharmaceutical Sciences, University of Trieste, Via Giorgieri 1, 34127 Trieste, Italy; antonella.calabretti@deams.units.it (A.C.);; 2Department of Life Sciences, University of Trieste, Via Valerio 28, 34127 Trieste, Italy; mromano@units.it; 3Italian Institute of Technology (IIT), Via E. Melen 83, 16152 Genova, Italy; sara.fortuna@iit.it; 4Department of Drug Sciences, Medicinal Chemistry and Pharmaceutical Technology Section, University of Pavia, Viale Taramelli 6 and 12, 27100 Pavia, Italy; simona.collina@unipv.it; 5Department of Drug and Health Sciences, University of Catania, Viale Doria 6, 95125 Catania, Italy; eamata@unict.it (E.A.);

**Keywords:** neuroprotective agents, sigma 1 receptor, acetylcholinesterase, antioxidant properties, docking

## Abstract

Neurodegeneration is a slow and progressive loss of neuronal cells or their function in specific regions of the brain or in the peripheral system. Among several causes responsible for the most common neurodegenerative diseases (NDDs), cholinergic/dopaminergic pathways, but also some endogenous receptors, are often involved. In this context, sigma 1 receptor (S1R) modulators can be used as neuroprotective and antiamnesic agents. Herein, we describe the identification of novel S1R ligands endowed with antioxidant properties, potentially useful as neuroprotective agents. We also computationally assessed how the most promising compounds might interact with the S1R protein’s binding sites. The in silico predicted ADME properties suggested that they could be able to cross the brain-blood-barrier (BBB), and to reach the targets. Finally, the observation that at least two novel ifenprodil analogues (**5d** and **5i**) induce an increase of the mRNA levels of the antioxidant NRF2 and SOD1 genes in SH-SY5Y cells suggests that they might be effective agents for protecting neurons against oxidative damage.

## 1. Introduction

Neurodegeneration (ND) is a common final pathway present in aging and neurodegenerative diseases (NDDs), which leads to irreversible neuronal damage and death [1]. NDDs are a heterogeneous group of disorders characterized by the progressive deterioration of the structure and function of cells and their networks in the central (CNS) and peripheral nervous system [2]. Alzheimer’s disease (AD), Parkinson’s disease (PD), Huntington disease (HD), amyotrophic lateral sclerosis (ALS), and multiple sclerosis (MS) are the most common NDDs and the second leading cause of death worldwide in 2016 [3]. The etiopathogenesis of NDD is quite heterogeneous, and structural alterations as well as pathologically altered proteins are associated with selective dysfunctions of neurotransmitter pathways (in particular acetylcholine and dopamine systems) and progressive loss of synapses and neurons [4,5,6].

Besides environmental and genetic factors, oxidative stress, that leads to overproduction of reactive oxygen species (ROS), can cause neuronal death as a result of alterations of different cellular targets, such as proteostasis, mitochondria, neurotransmitter metabolism or deregulation of antioxidant pathways [7,8,9].

Currently, treatments for NDDs are mostly symptomatic, and there is a compelling need for novel therapies that might be potentially able to change the course of the diseases.

The N-Methyl-D-Aspartate (NMDA) receptor containing GluN2b subunit (GluN2bR) [10,11] is essential as a control unit for the glutamatergic network in the central nervous system (CNS), keeping the excitatory neurotransmission balanced [12]. Overstimulation of NMDARs, as a consequence of (S)-glutamate surfeit and the subsequent uncontrolled neuronal influx of Ca^2+^ ions, induces excitotoxicity and triggers cell death by apoptosis.

This event is one of the main causes of the onset and worsening of several NDDs including AD, PD, HA, and others. Ifenprodil (**1**, Figure 1) [13] is the prototypical allosteric negative modulator that interacts selectively with the GluN2bR subunit blocking the agonistic excitatory effect of glutamate. In this contest, GluN2bR inhibitors are useful to antagonize the excitotoxicity in NDDs.

Sigma 1 receptors (S1R) also play a relevant role in neuroprotection. They show antiamnesic activities [14], and are involved in the modulation of opioid analgesia [15], schizophrenia without producing extrapyramidal side effects [16,17], and drug (cocaine) dependence [18]. Donepezil (Figure 2), one of the key drugs used in therapy for treating AD shows a high affinity for S1R (Ki = 14.6 nM) [19], although its main mechanism of action relies on inhibition of acetylcholinesterase (AChE), increasing the concentration of acetylcholine (ACh) at the synaptic level and thus restoring the cognitive functions in AD patients [20].

Based on these considerations, and in continuation of our efforts in discovering new SR modulators, we designed and synthesized new molecules structurally related to ifenprodil, the NMDAR modulator mentioned before, for which a moderate S1R affinity with KiS1 = 125 nM was also demonstrated [21].

Specifically, we designed molecules (**5a–o**) (Figure 1), by retaining the original phenylpropyl motif of ifenprodil, and by jointly replacing the 4-benzylpiperidine fragment with other cyclic or linear amines present in some well-known SR ligands, [22]. After in silico evaluating their drug-likeness and ability to bypass the blood-brain-barrier (BBB), all compounds have been successfully synthesized, properly characterized, and evaluated for affinity to both S1R and S2R through radioligand binding assay. Lastly, the in vitro antioxidant ability of the most interesting derivatives was evaluated as well as their potential interaction with the antioxidant response by upregulating the expression of SOD1 and NRF2 genes.

## 2. Results and Discussion

### 2.1. Compound Design

The new compounds have been designed by applying a hybridization approach, merging the key structural features of ifenprodil and donepezil. As shown in Figure 2, they present a benzylpiperazine (i.e., compound **5h**) or a benzyldiazepane moiety, which can mimic the benzylpiperidine fragment of the lead AChE inhibitor donepezil, and maintain the same length of the linker between the aromatic and the aminic portion (six carbons) which ensures a good compromise to drive the affinity towards S1R as well as AChE protein.

Before the synthesis, we in silico predicted the drug-likeness properties of the designed compounds and evaluated their ability to bypass the BBB, taking into account that the final aim of this work is to discover compounds active for CNS-related pathologies.

Using the SwissADME tool (www.swissadme.ch, accessed on 18 April 2022), we in silico evaluated all compounds for a prediction of the drug-likeness properties [23], with the most common pharmacokinetic parameters, on the basis of the extended version of Lipinski’s rule of five (RO5) [24]. The RO5 extended criterion means that an orally active drug should not violate more than one of the following requirements: MW ≤ 500; HBA and HBD (related to the membrane permeability) ≤10 and ≤5, respectively; logP and logS (related to the intestinal absorption) ≤ 5; PSA ≤ 140 Å. (Table 1).

All the evaluated compounds **1h** and **1k–n**, in comparison with haloperidol, ifenprodil, and donepezil as references standard, exhibited good drug-likeness properties being all the values within the ranges of RO5, suggesting that our new derivatives can penetrate the BBB and reach the targets.

### 2.2. Chemistry

The hybrid compounds were synthesized by alkylation of substituted amines, usually present in some SR ligands, with substituted 2-bromopropiophenones.

The synthetic route is depicted in Figure 1. It starts with the bromination of substituted propiophenones **2a–c** with CuBr_2_ to afford the corresponding 2-bromopropiophenones **3a–c** in good yield (68–94%), following a slight modification of a known procedure [25]. The latter was made to react with various amines **4a–d**, in basic media, to give the final products **5a–l**. All the amines were commercially available, with the exception of 1-benzylpiperazine, which was obtained from the reaction of benzyl chloride and excess of piperazine, in THF. Reduction of the carbonyl group of compounds **5c**, **5g,** and **5e** with NaBH_4_, affords the corresponding derivatives **5m–o**. All the compounds synthesized were properly characterized and all the spectra (Appendix A) were in agreement with the structure predicted.

### 2.3. Biology and Computational

#### 2.3.1. SR Binding Affinities, SAR Discussion, Molecular Dynamics and Docking Studies

The S1R and S2R receptor affinities of the test compounds were determined in competition experiments by radiometric assays. The collected affinity results for the new derivatives **5a–o** are reported in Table 2.

Among the three subseries, the best results were achieved by benzylpiperazine-based **5d**, **5h,** and **5k** and diazepane-based **5e**, **5i,** and **5o** derivatives. Compounds **5d** and **5h**, belonging to the piperazine-based series, showed also the best selective profile with S2/S1 ratios of 38 and 60, respectively. The 4-phenylpiperazine and N-methyl-4-phenylbutylamino fragments, led to moderate results towards both SR subtypes, probably due to the shorter or longer distance between the nitrogen basic atom and the phenyl residue, respectively. Conversely, the substitution with a methoxy or hydroxy group on the aromatic portion of the phenylpropanone fragment, generally increases the affinity for both SR subtypes, with the exception of the unsubstituted compound **5d** which retains a high affinity for S1R and a favorable selectivity ratio (KiS1 = 8.0 nM and S1/S2 = 38). Among the subseries **5m–o**, the best result was achieved by diazepane-based compound **5o**, which showed high S1R affinity (KiS1 = 4.2 nM), with a 35-fold higher selectivity towards S1R with respect to the S2R subtype. Generally, the free butylamino chain led to a worse result than the related cyclic amines (diazepane and piperazine nucleus).

Experimental results were confirmed by docking a subset of compounds to the S1R protein. For all molecules, the predicted binding affinities were compatible with those measured experimentally, thus suggesting that all molecules could be potential ligands for S1R, as can be appreciated for each compound by looking at the results associated with the two largest docking clusters (Table 3).

The optimum compound **5h** was further simulated for 20 ns by means of atomistic molecular dynamics (MD) simulation in full water solvent. Along the simulated time, **5h** did not leave its binding site, while it changed conformations as confirmed by visually comparing its initial conformation, output from the docking, to the final conformation obtained at the end of the MD trajectory (Figure 3a), as well as by tracking the root mean squared deviation (RMSD) of both protein and substrate along the simulated time (Figure 3b). Its binding free energy was re-estimated by means of MMGBSA calculations over the trajectory (Figure 3c) leading to a Kd of 2.5 nM. The energy decomposition of the free energy of binding shows that the substrate is kept in place primarily by van der Waals forces (Figure 3c). Indeed, in the pocket, the substrate is surrounded by 27 residues (Figure 3d). Of the 27 closest residues, 24 residues contribute to the binding with Ile124 and Thr 181 contributing more than 2 kcal/mol (Figure 3e). Asp126, Trp164, and Glu172 instead oppose the binding. Their unfavorable contribution to the binding energy is mainly due to their high polar solvation energy which is only partially counterbalanced by a gain in electrostatic and Van der Walls (Figure 3e). This latter energetic contribution is responsible for the high binding affinity of compound **5h**. An electrostatic contribution larger than 2 kcal/mol, other than those from Asp 126 and Glu172 include, that from Tyr120. Also, in this case, the polar solvation energy strongly opposes the binding, but here is counterbalanced by strong electrostatic and dispersion forces.

#### 2.3.2. Cytotoxic Profile

Before evaluating the antioxidant properties of compounds under investigation (**5e, 5i, 5d, 5o, 5h**, and **5k**), we tested their cytotoxicity on the human neuroblastoma SH-SY5Y cell line, a widely used neuronal model for similar studies [26,27]. The well-known S1R antagonists NE100 and Haloperidol and the S2R agonist Siramesine have been also tested for comparative purposes.

All compounds showed toxicity < 10% at 12.5 µM (curves are reported in Appendix A), with IC_50_ ranging between 74 and 294 µM (Table 4). The lowest cytotoxic effect was shown by derivative **5i**.

#### 2.3.3. Effects of the Novel Ifenprodil Analogues on Antioxidant SOD1 and NRF2 in SH-SY5Y Cells

Previous studies have shown that S1 agonists (such as pentazocine) seem to attenuate oxidative stress in the retinas of rd10mice [28,29] as well as in a zebrafish model ALS of TDP-43 pathology [30] possibly by increasing the expression of the nuclear erythroid 2-related factor 2 (NRF2), a basic leucine zipper transcription factor regulating the expression of over 500 antioxidant and cytoprotective genes. In addition, it was reported that S1 agonists trigger activation of the antioxidant response elements (ARE) and cause an increase of superoxide dismutase 1 (SOD1) mRNA expression in COS cells [31].

Based on these findings, we tested the neuroprotective properties of the novel analogs by evaluating their ability to activate an antioxidant response by upregulating the expression of SOD1 and NRF2 genes. To this aim, we monitored, by Real-Time PCR, whether incubation of SH-SY5Y cells with pentazocine, haloperidol, or the compounds **5e**, **5i**, **5d**, **5o**, **5h**, and **5k** was associated with an increase of the mRNA levels of SOD1 and NRF2 genes. Indeed, when cells were exposed to pentazocine for 12 h, the endogenous mRNA levels of SOD1 and NRF2 raised 1.3- and 1.8-fold, respectively, as compared to DMSO treatment (Figure 4a,b). Incubation with haloperidol did not cause significant changes in mRNA levels of both genes of interest instead. The novel compounds showed a different behavior, depending on their structure: incubation with **5d** was associated with a 1.3-fold increase in SOD1 mRNA levels, similar to what was observed with pentazocine.

Treatment with the compounds **5e** and **5i** did not significantly alter the gene expression, whereas the compounds **5o**, **5h**, and **5k** dramatically reduced the expression of SOD1 when compared to DMSO (Figure 4a).

Concerning the effects on NRF2 expression, whereas **5e** did not change its expression significantly, **5d** and **5i** led to a 1.7-fold and 2.8-fold increase in mRNA levels of the gene (compared to DMSO). Treatment with **5o**, **5h,** and **5k** was associated with a significant decrease in gene expression (Figure 4b).

Our findings further imply that analogs **5d** and **5i** can upregulate SOD1 and NRF2 expression in human SH-SY5Y cells in a manner similar to pentazocine, suggesting that they may be useful agents for preventing oxidative damage to neurons. Conversely, **5o**, **5h**, and **5k** have been discarded since they seem to suppress both SOD1 and NRF2 expression.

### 2.4. Antioxidant Activity

#### In Vitro Intrinsic Antioxidant Activity Evaluation

Total antioxidant activities of the most interesting compounds (**5d**, **5e**, **5h**, **5i, 5k,** and **5o**) were also evaluated by testing the ABTS (2,2′-azino-bis(3-ethylbenzothiazoline-6-sulphonic acid)) radical and hydrogen peroxide (H_2_O_2_) scavenging abilities. The synthetic antioxidant Trolox (6-hydroxy-2,5,7,8-tetramethylchroman-2-carboxylic acid) was used as a standard antioxidant reference. Four out of six compounds potently inhibited ABTS radicals and H_2_O_2_, compared to the standard (Table 5).

Compounds **5d**, **5e**, **5i,** and **5o**, exhibited a significant radical scavenging capacity both on the ABTS and H_2_O_2_, with IC_50_ values comparable to Trolox. All the diazepane-based compounds tested showed the best scavenging property.

## 3. Materials and Methods

### 3.1. Chemistry

#### 3.1.1. Chemical Reagents and Instruments

All reagents and solvents were of analytical grade and used as received. Flash chromatography was performed using Silica Gel 60 (70–230 mesh, Merck, Milan, Italy). Reaction courses were monitored on precoated silica gel TLC-GF_254_ plates (Merck) and spots were visualized under ultraviolet light at 254 nm or iodine vapors. Melting points (°C) were determined with a Stuart SMP 300 apparatus in open glass capillaries and were uncorrected. Agilent Cary-60 spectrophotometer UV-Vis was employed to record the spectra and quantify the absorbance. Infrared spectra were recorded on an FTIR Jasco 4700 spectrophotometer in nujol mulls. Nuclear magnetic resonance spectra were obtained on a Varian 400 MHz. Chemical shifts are reported as δ (ppm) in CDCl_3_ solution related to tetramethyl silane as an internal standard; 1 drop of D_2_O was added to assign NH or OH protons. ^1^H-^1^H coupling constants (*J*) are given in Hz and the splitting abbreviations used are s, singlet; d, doublet; dd, doublet of doublets; ddd: doublet of doublet of doublets; t, triplet; tt, triplet of triplets; dt, doublet of triplets; q, quartet; m, multiplet. Microanalyses (C, H, N) were carried out with Elementar Vario ELIII apparatus and were in agreement with theoretical values ±0.4%. ESI-MS spectra were recorded on a Bruker Daltonics Esquire 4000 spectrometer (MeOH ultrapure as solvent).

#### 3.1.2. Synthetic Procedure

##### General Synthesis of Brominated Compounds **3a–c**

*2-Bromo-1-phenylpropan-1-one* (**3a**)

CuBr_2_ (10.0 g 44.7 mmol, 2eq) was dissolved in 100 mL of EtOAc and heated at reflux temperature on a magnetic stirrer hot plate, then 3.0 g of propiophenone **2a** (22.4 mmol, 1 eq) was added. At reaction completion (the color of the solution changed from green to amber; TLC: CH_2_Cl_2_/EtOH 95:5) the inorganic salt of copper (I) bromide was collected by filtration and washed with fresh EtOAc. The organic phase was washed with water (3 × 100 mL) at neutrality, dried over MgSO_4_, filtered, and evaporated to dryness to give a chromatographically pure yellow oil **3a**.

Yield: 4.35 g, 91%. Rf: 0.79 (CH_2_Cl_2_/EtOH 95:5). ^1^H-NMR (400 MHz, CDCl_3_): δ 8.06–7.98 (m, 2H, arom.), 7.63–7.52 (m, 1H, arom.), 7.53–7.41 (m, 2H, arom.), 5.29 (q, *J* = 6.6 Hz, 1H, C*H*), 1.90 (d, *J* = 6.6 Hz, 3H, C*H*_3_). MS-ESI: [M+H]^+^ = 213, [M+H+2]^+^ = 215.

With the same procedure, but starting from 4-hydroxypropiophenone and 4-methoxypropiophenone, compounds **3b** and **3c** were obtained.

*2-Bromo-1-(4-methoxyphenyl)propan-1-one* (**3b**)

Whitish solid, yield: 2.02 g, 68%. Mp: 62–64 °C. Rf: 0.64 (CH_2_Cl_2_). ^1^H-NMR (400 MHz, CDCl_3_): δ 8.05–7.96 (m, 2H, arom.), 6.99–6.90 (m, 2H, arom.), 5.26 (q, *J* = 6.6 Hz, 1H, C*H*), 3.87 (s, 3H, OC*H*_3_), 1.88 (d, *J* = 6.6 Hz, 3H, C*H*_3_). MS-ESI: [M+H]^+^ = 243, [M+H+2]^+^ = 245.

*2-Bromo-1-(4-hydroxyphenyl)propan-1-one* (**3c**)

Light brown solid, yield: 2.86 g, 94%. Mp: 90–92 °C. Rf: 0.63 (CH_2_Cl_2_). ^1^H-NMR (400 MHz, CDCl_3_): δ 7.97 (d, *J* = 8.8 Hz, 2H, arom.), 6.92 (d, *J* = 8.9 Hz, 2H, arom.), 6.14 (s, 1H, OH), 5.25 (q, *J* = 6.6 Hz, 1H, C*H*), 1.89 (d, *J* = 6.6 Hz, 3H, C*H*_3_). MS-ESI: [M+H]^+^ = 229, [M+H+2]^+^ = 231.

##### General Synthesis of the Final Compounds **5a–m**

*2-(4-Benzyl-1,4-diazepan-1-yl)-1-phenylpropan-1-one* (**5a**)

On an ice bath (0 °C), a 100 mL round bottom flask with a mixture of 203 mg of compound **3a** (0.95 mmol, 1 eq), 181 mg (0.95 mmol, 1 eq) of *N*-benzylomopiperazine (1-benzyl-1,4-diazepane) **4a**, 198 mg (1.42 mmol, 1.5 eq) of K_2_CO_3_, a catalytic amount of KI and 50 mL of ACN, was left to stir overnight. At reaction completion, the inorganic salts were collected by filtration, and the organic phase was washed with distilled water (3 × 30 mL), dried over MgSO_4_, filtered, and evaporated under reduced pressure. No further purification was required.

Yellow oil, yield: 264 mg, 81%. FT-IR (cm^−1^): 1679. ^1^H-NMR (400 MHz, CDCl_3_): δ 8.10–8.03 (m, 2H, arom.), 7.60–7.37 (m, 3H, arom.), 7.38–7.16 (m, 3H, arom.), 4.27 (q, *J* = 6.7 Hz, 1H, C*H*), 3.55 (s, 2H, C*H*_2_), 2.89–2.71 (m, 4H, C*H*_2_, diazep.), 2.63–2.55 (m, 3H, C*H*_2_, diazep.), 2.46 (ddd, *J* = 12.8, 7.2, 3.2 Hz, 1H, C*H*_2_ diazep.), 1.80–1.60 (m, 2H, C*H*_2_ diazep.), 1.25 (d, *J* = 6.7 Hz, 3H, C*H*_3_). ^13^C-NMR (101 MHz, CDCl_3_): δ 200.89, 139.47, 136.59, 132.63, 128.88, 128.76, 128.17, 128.10, 126.75, 64.10, 62.02, 56.38, 53.89, 50.85, 50.57, 28.46, 10.18. MS-ESI: [M+H]^+^ = 323; elemental analysis calcd (%) for C_21_H_26_N_2_O: C 78.22, H 8.13, N 8.69; found: C 78.10, H 8.35, N 8.50.

In addition to *N*-benzylomopiperazine, 1-phenylpiperazine, *N*-methyl-4-phenylbutan-1-amine, and 1-benzylpiperazine were used to afford compounds **5b–l**.

*2-(4-Phenylpiperazin-1-yl)-1-phenylpropan-1-one* (**5b**)

Brown solid, yield: 228.4 mg, 81%. Mp: 105–106 °C. Rf: 0.41 (CH_2_Cl_2_). FT-IR (cm^−1^): 1679. ^1^H-NMR (400 MHz, CDCl_3_): δ 8.17–8.09 (m, 2H, arom.), 7.61–7.51 (m, 1H, arom.), 7.50–7.41 (m, 2H, arom.), 7.30–7.21 (m, 2H, arom.), 6.96–6.80 (m, 3H, arom.), 4.16 (q, *J* = 6.8 Hz, 1H, C*H*), 3.25–3.10 (m, 4H, C*H*_2_ pip.), 2.86–2.68 (m, 4H, C*H*_2_ pip.), 1.34 (d, *J* = 6.8 Hz, 3H, C*H*_3_). ^13^C-NMR (101 MHz, CDCl_3_): δ 200.34, 151.31, 136.26, 133.01, 129.06, 128.88, 128.40, 119.72, 116.10, 64.55, 49.63, 49.47, 11.64. MS-ESI: [M+H]^+^ = 295, [M+Na]^+^ = 317; elemental analysis calcd (%) for C_19_H_22_N_2_O: C 77.52, H 7.53, N 9.52; found: C 77.75, H 7.50, N 9.55.

*2-(Methyl(4-phenylbutyl)amino)-1-phenylpropan-1-one* (**5c**)

Yellow oil, yield: 230 mg, 80 %. Rf: 0.25 (CH_2_Cl_2_/EtOH 95:5). FT-IR (cm^−1^): 1677. ^1^H-NMR (400 MHz, CDCl_3_): δ 8.11–8.03 (m, 2H, arom.), 7.60–7.46 (m, 1H, arom.), 7.47–7.38 (m, 2H, arom.), 7.33–7.22 (m, 2H, arom.), 7.27–7.13 (m, 1H, arom.), 7.18–7.06 (m, 2H, arom.), 4.22 (q, *J* = 6.7 Hz, 1H, C*H*), 2.68–2.42 (m, 4H, NC*H*_2_CH_2_CH_2_C*H*_2_), 2.26 (s, 3H, NC*H*_3_), 1.64–1.42 (m, 4H, NCH_2_C*H*_2_C*H*_2_CH_2_), 1.25 (d, *J* = 6.7 Hz, 3H, C*H*_3_). ^13^C-NMR (101 MHz, CDCl_3_): δ 200.91, 142.49, 136.58, 132.69, 128.90, 128.40, 128.35, 128.34, 128.27, 128.20, 125.94, 125.61, 63.42, 53.68, 37.69, 35.66, 28.85, 28.59, 27.38, 9.40. MS-ESI: [M+H]^+^ = 296; elemental analysis calcd (%) for C_20_H_25_NO: C 81.31, H 8.53, N 4.74; found: C 81.45, H 8.33, N 4.62.

*2-(4-Benzylpiperazin-1-yl)-1-phenylpropan-1-one* (**5d**)

Yellow oil, yield: 182 mg, 80%. Rf: 0.21 (CH_2_Cl_2_/EtOH 95:5). FT-IR (cm^−1^): 1685. ^1^H-NMR (400 MHz, CDCl_3_): δ 8.13–8.05 (m, 2H, arom.), 7.62–7.39 (m, 3H, arom.), 7.35–7.19 (m, 5H, arom.), 4.09 (q, *J* = 6.8 Hz, 1H, C*H*), 3.50 (s, 2H, C*H*_2_), 2.78–2.56 (m, 4H, C*H*_2_ pip.), 2.48 (s, 4H, C*H*_2_ pip.), 1.28 (d, *J* = 6.8 Hz, 3H, C*H*_3_). ^13^C-NMR (101 MHz, CDCl_3_): δ 200.43, 136.28, 132.91, 129.28, 128.85, 128.83, 128.70, 128.66, 128.34, 128.22, 127.15, 109.99, 64.44, 62.91, 53.23, 49.41, 11.68. MS-ESI: [M+H]^+^ = 309; elemental analysis calcd (%) for C_20_H_24_N_2_O: C 77.89, H 7.84, N 9.08; found: C 77.70, H 7.90, N 9.05.

*2-(4-Benzyl-1,4-diazepan-1-yl)-1-(4-methoxyphenyl)propan-1-one* (**5e**)

Light brown oil, yield: 248 mg, 83%. Rf: 0.40 (CH_2_Cl_2_/EtOH 95:5). FT-IR (cm^−1^): 1675. ^1^H-NMR (400 MHz, CDCl_3_): δ 8.13–8.04 (m, 2H, arom.), 7.35–7.16 (m, 5H, arom.), 6.95–6.87 (m, 2H, arom.), 4.22 (q, *J* = 6.7 Hz, 1H, C*H*), 3.86 (d, *J* = 0.6 Hz, 3H, OC*H*_3_), 3.57 (s, 2H, C*H*_2_), 2.89–2.68 (m, 4H, C*H*_2_, diazep.), 2.70–2.56 (m, 3H, C*H*_2_, diazep.), 2.48 (ddd, *J* = 12.7, 6.8, 3.6 Hz, 1H, C*H*_2_, diazep.), 1.81–1.61 (m, 2H, C*H*_2_, diazep.), 1.24 (d, *J* = 6.7 Hz, 3H, C*H*_3_). ^13^C-NMR (101 MHz, CDCl_3_): δ 199.41, 163.12, 139.49, 131.25, 129.44, 128.77, 128.09, 126.75, 113.62, 113.31, 63.98, 62.09, 56.43, 55.38, 53.90, 50.85, 50.61, 28.46, 10.39. MS-ESI: [M+H]^+^= 353, [M+Na]^+^ = 375; elemental analysis calcd (%) for C_22_H_28_N_2_O_2_: C 74.97, H 8.01, N 7.95; found: C 74.70, H 8.15, N 8.05.

*2-(4-Phenylpiperazin-1-yl)-1-(4-methoxyphenyl)propan-1-one* (**5f**)

Light brown solid, yield: 230 mg, 86%; Mp: 117–119 °C. Rf: 0.42 (CH_2_Cl_2_/EtOH 95:5) FT-IR (cm^−1^): 1680. ^1^H-NMR (400 MHz, CDCl_3_): δ 8.19–8.08 (m, 2H, arom.), 7.31–7.19 (m, 2H, arom.), 6.98–6.79 (m, 5H, arom.), 4.08 (q, *J* = 6.8 Hz, 1H, *CH*), 3.87 (s, 3H, OC*H*_3_), 3.29–3.09 (m, 4H, CH_2_ pip.), 2.85–2.66 (m, 4H, C*H*_2_ pip.), 1.33 (d, *J* = 6.8 Hz, 3H, C*H*_3_). ^13^C-NMR (101 MHz, CDCl_3_): δ 198.94, 163.42, 151.32, 131.26, 129.13, 129.05, 119.68, 116.07, 113.53, 64.62, 55.44, 49.72, 49.45, 12.08. MS-ESI: [M+H]^+^ = 325; elemental analysis calcd (%) for C_20_H_24_N_2_O_2_: C 74.05, H 7.46, N 8.63; found: C 74.00, H 7.52, N 8.57.

*1-(4-Methoxyphenyl)-2-(methyl(4-phenylbutyl)amino)propan-1-one* (**5g**)

Yellow oil, yield: 259 mg, 94%. Rf: 0.29 (CH_2_Cl_2_/EtOH 95:5). FT-IR (cm^−1^): 1675. ^1^H-NMR (400 MHz, CDCl_3_): δ 8.13–8.05 (m, 2H, arom.), 7.30–7.06 (m, 5H, arom.), 6.99–6.83 (m, 2H, arom.), 4.17 (q, *J* = 6.7 Hz, 1H, C*H*), 3.82 (s, 3H, OC*H*_3_*),* 2.69–2.43 (m, 4H, NC*H*_2_CH_2_CH_2_C*H*_2_), 2.26 (s, 3H, NC*H*_3_), 1.62–1.44 (m, 4H, NCH_2_C*H*_2_C*H*_2_CH_2_), 1.24 (d, *J* = 6.7 Hz, 3H, C*H*_3_). ^13^C-NMR (101 MHz, CDCl_3_): δ 199.34, 163.23, 142.51, 133.93, 131.29, 131.03, 129.37, 128.34, 128.26, 128.20, 125.61, 113.43, 113.22, 63.44, 55.48, 55.35, 53.63, 37.85, 37.69, 35.64, 28.84, 27.33, 9.63. MS-ESI: [M+H]^+^ = 326; elemental analysis calcd (%) for C_21_H_27_NO_2_: C 77.50, H 8.36, N 4.30; found: C 77.55, H 8.30, N 4.25.

*2-(4-Benzylpiperazin-1-yl)-1-(4-methoxyphenyl)propan-1-one* (**5h**)

Brownish solid, yield: 138 mg, 69%. Mp: 80–82 °C. Rf: 0.42 (CH_2_Cl_2_/EtOH 95:5). FT-IR (cm^−1^): 1675. ^1^H-NMR (400 MHz, CDCl_3_): δ 8.15–8.07 (m, 2H, arom.), 7.30 (d, *J* = 4.4 Hz, 4H, arom.), 7.27–7.20 (m, 1H, arom.), 6.99–6.88 (m, 2H, arom.), 4.01 (q, *J* = 6.7 Hz, 1H, C*H*), 3.87 (s, 3H, OC*H*_3_), 3.49 (s, 2H, C*H*_2_), 2.70–2.61 (m, 2H, C*H*_2_ pip.), 2.59 (m, 2H, C*H*_2_ pip.), 2.49 (d, *J* = 12.4 Hz, 4H, C*H*_2_ pip.), 1.27 (d, *J* = 6.8 Hz, 3H, C*H*_3_). ^13^C-NMR (101 MHz, CDCl_3_): δ 199.06, 163.33, 131.28, 131.21, 129.23, 129.21, 128.19, 127.07, 113.94, 113.46, 109.99, 64.53, 62.95, 55.41, 53.27, 12.09. MS-ESI: [M+H]^+^ = 339; elemental analysis calcd (%) for C_21_H_26_N_2_O_2_: C 74.53, H 7.74, N 8.28; found: C 74.65, H 7.55, N 8.18

*2-(4-Benzyl-1,4-diazepan-1-yl)-1-(4-hydroxyphenyl)propan-1-one* (**5i**)

Light brown oil, yield: 228 mg, 74 %. Rf: 0.15 (CH_2_Cl_2_). FT-IR (cm^−1^): 3332, 1652. ^1^H-NMR (400 MHz, CDCl_3_): δ 7.92 (d, *J* = 8.6 Hz, 2H, arom.), 7.28–7.21 (m, 5H, arom.), 7.15 (s all, 1H, O*H*), 6.77 (d, *J* = 8.5 Hz, 2H, arom.), 4.24 (q, *J* = 6.7 Hz, 1H, C*H*), 3.63 (s, 2H, C*H*_2_), 2.85 (ddd, *J* = 18.0, 8.0, 3.5 Hz, 4H, C*H*_2_, diazep.), 2.71 (dt, *J* = 10.1, 5.3 Hz, 3H, C*H*_2_, diazep.), 2.70–2.54 (m, 1H, C*H*_2_, diazep.), 1.86–1.75 (m, 2H, C*H*_2_, diazep.), 1.23 (d, *J* = 6.7 Hz, 3H, C*H*_3_). ^13^C-NMR (101 MHz, CDCl_3_): δ 199.58, 162.36, 137.45, 131.45, 129.39, 128.31, 127.82, 127.33, 115.66, 63.23, 61.83, 55.80, 53.62, 50.72, 49.38, 27.31, 11.40. MS-ESI: [M+H]^+^ = 339; elemental analysis calcd (%) for C_21_H_26_N_2_O_2_: C 74.53, H 7.74, N 8.28; found: C 74.30, H 7.85, N 8.35.

*1-(4-Hydroxyphenyl)-2-(methyl(4-phenylbutyl)amino)propan-1-one* (**5j**)

Light brown oil, yield: 198 mg, 70 %. Rf: 0.50 (CH_2_Cl_2_/EtOH 90:10). FT-IR (cm^−1^): 3330, 1660. ^1^H-NMR (400 MHz, CDCl_3_): δ 7.97 (dq, *J* = 8.9, 3.5, 3.1 Hz, 2H, arom.), 7.30–7.05 (m, 5H, arom.), 6.92–6.75 (m, 2H, arom.), 5.53 (s, 1H, O*H*), 4.23–4.07 (m, 1H, C*H*), 2.62–2.43 (m, 4H, NC*H*_2_CH_2_CH_2_C*H*_2_), 2.28 (s, 3H, NC*H*_3_), 1.78–1.43 (m, 4H, NCH_2_C*H*_2_C*H*_2_CH_2_), 1.28–1.16 (m, 3H, C*H*_3_). ^13^C-NMR (101 MHz, CDCl_3_): δ 199.69, 142.40, 131.50, 128.34, 128.32, 128.20, 125.85, 125.60, 115.42, 62.92, 53.90, 37.85, 35.62, 28.93, 27.05, 10.94. MS-ESI: [M+H]^+^ = 312; elemental analysis calcd (%) for C_20_H_25_NO_2_: C 77.14, H 8.09, N 4.50; found: C 77.05, H 7.93, N 4.55.

*2-(4-Benzylpiperazin-1-yl)-1-(4-hydroxyphenyl)propan-1-one* (**5k**)

Whitish solid, yield: 151 mg, 71%. Mp: 58–59 °C. Rf: 0.26 (CH_2_Cl_2_/EtOH 95:5). FT-IR (cm^−1^): 3325, 1672. ^1^H-NMR (400 MHz, CDCl_3_): δ 8.03–7.87 (m, 2H, arom.), 7.34–7.20 (m, 5H, arom.), 6.90–6.72 (m, 2H, arom.), 4.06–3.91 (m, 1H, C*H*), 3.56–3.48 (m, 2H, C*H*_2_), 2.69–2.43 (m, 8H, C*H*_2_ pip.), 1.30–1.19 (m, 3H, C*H*_3_). ^13^C-NMR (101 MHz, CDCl_3_): δ 198.99, 161.01, 131.45, 131.34, 129.57, 129.47, 128.62, 128.30, 127.40, 115.45, 63.83, 62.85, 53.00, 52.58, 12.98. MS-ESI: [M+H]^+^ = 325; elemental analysis calcd (%) for C_20_H_24_N_2_O_2_: C 74.05, H 7.46, N 8.63; found: C 74.20, H 7.35, N 8.44.

*2-(4-Phenylpiperazin-1-yl)-1-(4-hydroxyphenyl)propan-1-one* (**5l**)

Purified through *flash chromatography* (CH_2_Cl_2_/EtOH 98:2). Brown solid, yield: 83.0 mg, 47%. Mp: 163–165 °C. Rf: 0.75 (CH_2_Cl_2_/EtOH 95:5). FT-IR (cm^−1^): 3338, 1655. ^1^H-NMR (400 MHz, CDCl_3_): δ 8.11–8.02 (m, 2H, arom.), 7.35–7.19 (m, 2H, arom.), 6.95–6.80 (m, 5H, arom.), 4.11 (q, *J* = 6.8 Hz, 1H, C*H*), 3.20 (dt, *J* = 6.7, 3.5 Hz, 4H, C*H*_2_ pip.), 2.83 (dt, *J* = 10.6, 4.9 Hz, 2H, C*H*_2_ pip.), 2.74 (dt, *J* = 10.9, 5.0 Hz, 2H, C*H*_2_ pip.), 1.34 (d, *J* = 6.8 Hz, 3H, C*H*_3_). ^13^C-NMR (101 MHz, CDCl_3_): δ 198.96, 160.37, 151.15, 131.54, 129.08, 119.90, 116.19, 115.29, 64.43, 49.78, 49.37, 12.50. MS-ESI: [M+H]^+^ = 311; elemental analysis calcd (%) for C_19_H_22_N_2_O_2_: C 73.52, H 7.14, N 9.03; found: C 73.45, H 7.25, N 9.15.

##### General Synthesis for the Reduced Compounds **5m–o**

*2-(Methyl(4-phenylbutyl)amino)-1-phenylpropan-1-ol* (**5m**)

To a stirred solution of *2-(methyl(4-phenylbutyl)amino)-1-phenylpropan-1-one* **5c** (100 mg, 0.34 mmol, 1 eq) in abs, EtOH (20 mL) at 0 °C and 25.2 mg of NaBH_4_ (0.68 mmol, 2eq) were added portion-wise. The reaction mixture was allowed to reach room temperature and stirred for a further 8 h. The solvent was removed under reduced pressure, and water was added to the residue. The aqueous layer was extracted with CH_2_Cl_2_ (3 × 20 mL). The combined organic layers were dried (MgSO_4_), filtered, and concentrated in a vacuum to obtain **5m** as a yellow oil, chromatographically pure.

Yield: 75 mg, 74%. Rf: 0.54 (CH_2_Cl_2_/EtOH 90:10). FT-IR (cm^−1^): 3389. ^1^H-NMR (400 MHz, CDCl_3_): δ 7.33–7.15 (m, 10H, arom.), 4.81 (d, *J* = 4.4 Hz, 1H, C*H*OH), 2.80 (qd, *J* = 6.9, 4.0 Hz, 1H, C*H*CH_3_), 2.62 (t, *J* = 7.4 Hz, 2H, NCH_2_CH_2_CH_2_C*H*_2_), 2.51–2.31 (m, 2H, NC*H*_2_CH_2_CH_2_CH_2_), 2.24 (s, 3H, NC*H*_3_), 1.67–1.44 (m, 4H, NCH_2_C*H*_2_C*H*_2_CH_2_), 1.27 (s, 1H, O*H*), 0.86 (d, *J* = 6.8 Hz, 3H, C*H*_3_). ^13^C-NMR (101 MHz, CDCl_3_): δ 142.41, 142.32, 128.37, 128.28, 127.90, 126.80, 126.06, 125.70, 72.87, 63.52, 54.73, 38.84, 35.81, 29.70, 29.01, 27.01, 10.02. MS-ESI: [M+H]^+^ = 298; elemental analysis calcd (%) for C_20_H_27_NO: C 80.76, H 9.15, N 4.71; found: C 80.85, H 9.03, N 4.66.

The same procedure was adopted to obtain derivatives **5n,o**.

*1-(4-Methoxyphenyl)-2-(methyl(4-phenylbutyl)amino)propan-1-ol* (**5n**)

Dark yellow oil, yield: 32.4 mg, 32%. Rf: 0.59 (CH_2_Cl_2_/EtOH 90:10). FT-IR (cm^−1^): 3389. ^1^H-NMR (400 MHz, CDCl_3_): δ 7.34–7.21 (m, 4H, arom.), 7.26–7.14 (m, 4H, arom.), 6.93–6.81 (m, 2H, arom.), 4.17 (d, *J* = 9.7 Hz, 1H, C*H*OH), 3.80 (s, 3H, OC*H*_3_), 2.70–2.34 (m, 4H, NC*H*_2_CH_2_CH_2_C*H*_2_), 2.23 (s, 3H, NC*H*_3_), 1.76–1.50 (m, 4H, NCH_2_C*H*_2_C*H*_2_CH_2_), 1.34–1.24 (m, 1H, O*H*), 0.87 and 0.71 (d, *J* = 6.8 Hz, 3H, C*H*_3_). ^13^C-NMR (101 MHz, CDCl_3_): δ 159.14, 142.40, 134.39, 134.21, 128.41, 128.29, 127.16, 125.71, 113.62, 113.34, 74.19, 65.58, 55.25, 38.81, 35.81, 29.03, 27.78, 26.99, 7.19. MS-ESI: [M+H]^+^ = 328, [M+Na]^+^ = 350; elemental analysis calcd (%) for C_21_H_29_NO_2_: C 77.02, H 8.93, N 4.28; found: C 77.05, H 8.83, N 4.15.

*2-(4-Benzyl-1,4-diazepan-1-yl)-1-(4-methoxyphenyl)propan-1-ol* (**5o**)

Yellow oil, yield: 28.4 mg, 26%. Rf: 0.26 (CH_2_Cl_2_/EtOH 95:5). FT-IR (cm^−1^): 3389. ^1^H-NMR (400 MHz, CDCl_3_): δ 7.39–7.23 (m, 5H, arom.), 7.28–7.16 (m, 2H, arom.), 6.90–6.83 (m, 2H, arom.), 4.14 (d, *J* = 9.7 Hz, 1H, C*H*OH), 3.80 (s, 3H, OC*H*_3_), 3.67 (s, 2H, C*H*_2_), 2.94–2.57 (m, 9H, 4× C*H*_2_ diazep. and C*H*CH_3_), 1.87 (tt, *J* = 12.2, 10.7, 4.4 Hz, 2H, C*H*_2_ diazep.), 1.25 (s, 1H, O*H*), 0.76 (d, *J* = 6.6 Hz, 3H, C*H*_3_). ^13^C-NMR (101 MHz, CDCl_3_): δ 159.14, 133.94, 128.83, 128.40, 128.21, 126.92, 113.63, 74.38, 67.70, 62.78, 56.76, 55.24, 53.95, 28.46, 8.45. MS-ESI: [M+H]^+^ = 355; elemental analysis calcd (%) for C_22_H_30_N_2_O_2_: C 74.54, H 8.53, N 7.90; found: C 74.25, H 8.50, N 7.95.

### 3.2. Computational

#### 3.2.1. Docking

S1 protein was modeled by homology by using as a template chain A from PDB 5HK2 [32]. Details on the followed procedure can be found in [22]. Ligands were minimized with the AM1 method with MOPAC [33] and prepared with AutoDock tools. Docking was performed with AutoDock [34]. The docking box was centered on the ligand cocrystallized in 5HK2. A grid of 40 × 42 × 40 points with spacing 0.375 Å was employed. The docking was based on the Lamarckian Genetic Algorithm with 1000 runs and 2,500,000 maximum numbers of evaluations. The representative conformation of the largest cluster was selected for subsequent analysis.

#### 3.2.2. Molecular Dynamics

The minimization of the complex was performed in subsequent steps by constraining the selected portion of the systems. First side chains were minimized, then the whole protein, then the whole system. The system was then placed in a cubic box with periodic boundary conditions, and a 0.7 nm water layer was added before performing a further minimization. We used AMBER99SB force field and tip3p water. Ligand topologies were built with Antechamber [35] and converted into GROMACS topologies [36]. The 100 ps long NVT and NPT equilibrations were performed by constraining the protein backbone. NPT production runs of 20 ns of the unconstrained system were run at 300 K and 1 atm. A modified Berendsen thermostat and Parinello–Rahman pressure coupling were employed. The iteration time step was set to 2 fs with the leap-frog integrator and LINCS [37] constraint. Sampled conformations were clustered with the Daura algorithm [38] cutoff of 0.05 nm. RMSDs have been calculated from configurations sampled every 10 ps. All simulations and their analyses were performed with the Gromacs package v. 2021 [39]. The binding free energy was estimated with the MM/GBSA method by using the gmx_MMPBSA tool [40]. Apolar solvation energies were calculated as solvent-accessible surface area (SASA). All calculations were run on M100 (CINECA, Bologna, Italy).

### 3.3. Hydrogen Peroxide Radical Scavenging Activity

The compounds’ antioxidant capacity was evaluated by bleaching the green colored ethanolic solution of ABTS [41]. Test compounds (300 μL) were diluted as follows: 0.01, 0.025, 0.05, 0.1, and 0.2 mg/mL and added to 2.7 mL of ethanolic solution of ABTS (7 mM). These mixtures were incubated for 45 min at room temperature, and the absorbances were recorded at λ = 735 nm against the ABTS solution. The results were obtained as the percent of inhibition (% IC) of ABTS radical, calculated by the following formula.
% IC = [(Abs ABTS − Abs Sample)/Abs ABTS] × 100

Data were expressed as mean value ± SD, and the assay was performed in triplicate.

The % IC was used to determine the IC_50_ values.

The ABTS method was applied also to measure the IC_50_ of H_2_O_2_ (2 mM, λ = 230 nm), used as an oxidant compound comparing values.

### 3.4. Biology

#### 3.4.1. S1R and S2R Binding Assays

##### Materials

Brain and liver homogenates for S1R and S2R receptor binding assays were prepared from male Dunkin-Hartley guinea pigs and Sprague Dawley rats, respectively (ENVIGO RMS S.R.L., Udine, Italy; Italian Minister of Health, authorization for animal experimentation—Project acronym 335/1984F.N.JLT). [^3^H](+)-pentazocine (26.9 Ci/mmol) and [^3^H]1,3-di-*o*-tolylguanidine ([^3^H]DTG, 35.5 Ci/mmol) were purchased from PerkinElmer (Zaventem, Belgium). Ultima Gold MV Scintillation cocktail was from PerkinElmer (Milan, Italy). All the other materials were obtained from Merck Life Science S.r.l. (Milan, Italy). UV absorbance was measured using a microplate spectrophotometer reader (Synergy HT, Biotec). The bound radioactivity has been determined using a Beckman LS 6500 liquid scintillation counter (Beckman Coulter, Brea, CA, USA).

##### Preparation of the Test Compounds

The test compound solutions were prepared by dissolving approximately 10 µmol of the test compound in DMSO to obtain a 10 mM stock solution. The required final concentrations for the assay (from 10^−5^ to 10^−11^ M) have been reached by diluting the DMSO stock solution with the respective assay buffer.

##### Preparation of the Membranes from Pig Brain

Fresh guinea pig brain cortex was homogenized in ice-cold Tris (50 mM, pH 7.4) containing cold 0.32 M sucrose with a Potter-Elvehjem glass homogenizer. The suspension was centrifuged at 1030× *g* for 10 min at 4 °C. The supernatant was separated and centrifuged at 41,200× *g* for 20 min at 4 °C. The obtained pellet was suspended in ice-cold Tris (50 mM, pH 7.4), incubated at room temperature for 15 min, and centrifuged again at 41,200× *g* for 15 min at 4 °C. The final pellet was resuspended in ice-cold Tris buffer, and frozen at −80 °C. The protein concentration was determined by the method of Bradford [42].

##### Preparation of the Membranes from Rat Liver

A few rat livers were homogenized in cold 0.32 M sucrose with a Potter-Elvehjem glass homogenizer. The suspension was centrifuged at 1030× *g* for 10 min at 4 °C. The supernatant was separated and centrifuged at 31,100× *g* for 20 min at 4 °C. The pellet was resuspended in ice-cold Tris buffer (50 mM, pH 8) and incubated at room temperature for 30 min. Then, the suspension was centrifuged again at 31,100× *g* for 20 min at 4 °C. The final pellet was resuspended in ice-cold Tris buffer and stored at −80 °C. The protein concentration was determined by the method of Bradford.

##### S2R Ligand Binding Assay

In vitro S1R ligand binding assays were performed with [^3^H](+)-pentazocine (26.9 Ci/mmol). Guinea pig brain cortex homogenates (250 μg/sample) were incubated with increasing concentrations of test compounds, [^3^H](+)-pentazocine (2 nM) and Tris buffer (50 mM, pH 7.4) in a final volume of 0.5 mL, at 37 °C. Unlabeled (+)-pentazocine (10 μM) was used to measure non-specific binding. The K_d_ value of [^3^H](+)-pentazocine is 2.9 nM. Bound and free radioligand were separated by fast filtration under reduced pressure using a Millipore filter apparatus through Whatman GF 6 glass fiber filters, which were presoaked in a 0.5% poly(ethyleneimine) water solution for 120 min. Each filter paper was rinsed three times with 3 mL of ice-cold Tris buffer (50 mM, pH 7.4), dried at room temperature, and incubated overnight with 3 mL of Ultima Gold MV Scintillation cocktail into pony vials.

##### S2R Ligand Binding Assay

In vitro S2R ligand binding assays were carried out with [^3^H]1,3-di-*o*-tolylguanidine ([^3^H]DTG, 35.5 Ci/mmol). The thawed membrane preparation of rat liver (250 μg/sample) was incubated with increasing concentrations of test compounds, [^3^H]DTG (2 nM) in the presence of (+)-pentazocine (5 µM) as S1R masking agent, and Tris buffer (50 mM, pH 8.0) in a final volume of 0.5 mL, at room temperature. Non-specific binding was evaluated with unlabeled DTG (10 μM). The K_d_ value of [^3^H]DTG is 17.9 nM. Bound and free radioligand were separated by fast filtration under reduced pressure using a Millipore filter apparatus through Whatman GF 6 glass fiber filters, which were presoaked in a 0.5% poly(ethyleneimine) water solution for 120 min. Each filter paper was rinsed three times with 3 mL of ice-cold Tris buffer (10 mM, pH 8), dried at room temperature, and incubated overnight with 3 mL of scintillation fluid into pony vials.

##### Data Analysis

The Ki-values were calculated with the program GraphPad Prism^®^ 6.0 (GraphPad Software, San Diego, CA, USA). The Ki-values are given as the mean value ± SD from at least two independent experiments performed in duplicate.

#### 3.4.2. Cytotoxicity Studies

##### Cell Culture

The human neuroblastoma SH-SY5Y (ATCC CRL-2266) cells were maintained in Dulbecco’s modified Eagle’s medium (DMEM; Gibco, Life Technologies Inc., Frederick, MD, USA) supplemented with 10% (*v*/*v*) fetal bovine serum (FBS) and antibiotic antimycotic solution (100 U penicillin, 100 μg/mL streptomycin and 0.25 μg/mL amphotericin B; Sigma-Aldrich, St. Louis, MO, USA) at 37 °C in a humidified incubator with a 5% CO_2_/95% air atmosphere.

##### Cell Viability Test

Cell viability was tested by using Resazurin (SERVA Electrophoresis GmbH, Heidelberg, Germany), which in metabolically active cells is reduced to Resufurin, a redox indicator, whose fluorescence is proportional to the number of live cells [43,44]. The stock solution of resazurin sodium salt (440 μM, 10×) diluted in phosphate buffered saline (PBS) was prepared and stored at −20 °C. Working solution (44 μM resazurin) was prepared on the same day of each assay by diluting resazurin stock solution 1:10 in a standard culture medium. After the removal of residual media, 100 µL of a working solution containing 44 µM resazurin diluted in culture medium was added to each well of a p96 black plate.

Cells (5 × 10^3^), seeded in a black 96-well (clear bottom) plate, were incubated at 37 °C with 44 µM Resazurin solution equal to 10% of the complete medium volume (100 µL DMEM with 10% FCS + 11 µL Resazurin 440 µM).

The fluorescent signal of the resorufin was monitored with excitation λ = 530 nm and emission λ = 590 nm by using an Envision 2104 multi-label microplate reader (Perkin Elmer, Boston, MA, USA. Measurements were performed at time 0 and after 1 h of incubation.

The percentage of viability was calculated after subtraction of the background (obtained by killing the cells), on the basis of the ratio between the fluorescence values of the cells incubated with a compound and the fluorescence values of cells incubated with the solvent (1% DMSO). The viability of cells incubated with 1% DMSO was considered to be equivalent to 100%.

##### Data Analysis

The statistical analysis was carried out with GraphPad Prism^®^ 6.0 (GraphPad Software, Inc, La Jolla, CA, USA) software using an unpaired *t*-test. *p* < 0.05 was considered statistically significant. Cytotoxicity concentrations (IC_50s_) were determined from dose-response curves analyzed by using GraphPad Prism software.

##### Real-Time PCR to Test Neuroprotective Properties

Gene expression levels of genes known to be implicated in anti-oxidant response (SOD1 and NRF2) were determined by quantitative Real-Time PCR. SH-SY5Y cells (100.000/well) were seeded in 35 mm plates and incubated for 12 h with the **5e**, **5i**, **5d**, **5o**, **5h**, and **5k**, pentazocine, haloperidol, (12.5 µM) or DMSO (1%) for 12 h. Total RNA was extracted by using Trifast reagent (Euroclone, Milan, Italy), according to the manufacturer’s instructions.

Total RNA (500 ng) was primed with hexameric random primers and retrotranscribed into cDNA with M-MLV reverse transcriptase (Thermo Fisher Scientific, Waltham, MA, USA). The normalization of targets’ gene expression was performed against the housekeeping gene RPL13A. Real-Time PCRs were performed by using iQ SYBR Green Supermix and CFX96 Real-Time PCR detection system (Bio-Rad Laboratories, Redmond, WA, USA). The relative expression levels were calculated using the 2^−∆∆CT^ method. An unpaired T-test was used to determine statistical significance. Values are presented as mean and error bars indicate standard errors (SE). The results are representative of two independent experiments in triplicate.

Primer pair sequences were the following (5 to 3’): RPL13A_Ex7_Fw, CCTGGAGGAGAAGAGGAAAGAGA and RPL13A_Ex8_Rv, TTGAGGACCTCTGTGTATTTGTCAA; hSOD1_ex1_as, TTGCGTCGTAGTCTCCTG, and hSOD1_ex2_as, CACCTTCACTGGTCCATTAC; NFE2L2_Ex1for, ATCATGATGGACTTGGAGCTG; NFE2L2_Ex2rev, GCTCATACTCTTTCCGTCGC.

## 4. Conclusions

In order to find novel SR modulators with potential neuroprotective effects, we designed, synthesized, and in vitro tested novel ifenprodil analogs, a prototypical GluN2B receptor inhibitor. The synthesized compounds showed a preferential affinity for the S1R subtype, and 6 out of 15 derivatives have Ki S1R values <20 nM, with the piperazine derivative **5h** achieving the highest results (Ki S1R = 1.4 nM, selectivity over S2R = 60).

According to a preliminary in vitro antioxidant assay, four of the best S1R ligands have a greater intrinsic ability to scavenge ABTS-derived radicals and H_2_O_2_ than Trolox, the reference standard, and have a negligible impact on neuroblastoma SH-SY5Y cell viability.

Our results revealed that two derivatives of the series (**5d** and **5i**) induce an increase in the mRNA levels of the antioxidant NRF2 and SOD1 genes in SH-SY5Y cells, suggesting that they may be useful agents for preventing neurons from suffering from oxidative damage.

Although the precise mechanisms by which S1R can regulate SOD1 and NRF2 expression are not clear, considering that it plays a role in various cellular processes, such as calcium signaling, cellsurvival, and stress response [45,46], we hypothesize that its stimulation by agonists might regulate gene expression (with neuroprotective effects) through its known translocation to the endoplasmic reticulum, IP3 receptor stabilization, ER stress reduction, and modulation of calcium signaling, such as shown in several other cases [26,47,48,49].

## Data Availability

The data presented in this study are available on request from the corresponding author.

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
