# Peer review of "Cytotoxicity Profiles and Neuroprotective Properties of the Novel Ifenprodil Analogues as Sigma Ligands"

_molecules, 2023, doi:10.3390/molecules28083431_

Round 1

Reviewer 1 Report

This manuscript reports a series of small molecule drugs targeting S1R as potential therapeutics for neurodegenerative disease. The data showed that the physicochemical properties of these small molecules are well within the statistical range of CNS drugs. According to the results of binding and functional experiments, the effects of the molecules were also improved compared with the reference compounds. My only question is what the authors think the next optimization action would be. For instance, even the most selective molecule in this series has an S1R/S2R of less than 100-fold. Would it be a concern for the toxicity of those lead compounds?

Reviewer 2 Report

Please see review attached

Reviewer 3 Report

The manuscript of Zampieri et al. aims to identify new potential sigma 1 receptor (S1R ) ligand molecules with antioxidant and neuroprotective effects. Thus, they synthesize suitably substituted molecules and characterize them by evaluating the affinity for the S1R and S2R receptors through binding measurements as well as their characteristics in term of ability to cross the artificial BBB and antioxidant properties.

Overall, the manuscript is well organized and very clearly written. However, I have to ask  for same clarifications:

1) Fig 4A does not report whether the increase in PTZ and 5d expression is statistically significant although it is reported in the text that it is significant; also in line 221-222 correct the effect exerted by compund 5i; its role  is not clear.

2) the authors evaluate the antioxidant effects by means an in vitro assay with ABTS despite having shown the viability in SHSY5Y cells. It would also have been appropriate to evaluate the antioxidant action in the cell model as it is palusible that the effects could be significantly different. Could the authors explain?

change  "Cpd" in table 5 with "Cmpd" as other tables.
